# Complete Chloroplast Genome Sequences of *Kaempferia Galanga* and *Kaempferia Elegans*: Molecular Structures and Comparative Analysis

**DOI:** 10.3390/molecules24030474

**Published:** 2019-01-29

**Authors:** Dong-Mei Li, Chao-Yi Zhao, Xiao-Fei Liu

**Affiliations:** Guangdong Key Lab of Ornamental Plant Germplasm Innovation and Utilization, Environmental Horticulture Research Institute, Guangdong Academy of Agricultural Sciences, Guangzhou 510640, China; zhaoc2009@126.com (C.-Y.Z.); 13580525912@139.com (X.-F.L.)

**Keywords:** *Kaempferia galanga*, *Kaempferia elegans*, chloroplast genome, Illumina sequencing, PacBio sequencing, genomic structure, comparative analysis

## Abstract

*Kaempferia galanga* and *Kaempferia elegans*, which belong to the genus *Kaempferia* family Zingiberaceae, are used as valuable herbal medicine and ornamental plants, respectively. The chloroplast genomes have been used for molecular markers, species identification and phylogenetic studies. In this study, the complete chloroplast genome sequences of *K. galanga* and *K. elegans* are reported. Results show that the complete chloroplast genome of *K. galanga* is 163,811 bp long, having a quadripartite structure with large single copy (LSC) of 88,405 bp and a small single copy (SSC) of 15,812 bp separated by inverted repeats (IRs) of 29,797 bp. Similarly, the complete chloroplast genome of *K. elegans* is 163,555 bp long, having a quadripartite structure in which IRs of 29,773 bp length separates 88,020 bp of LSC and 15,989 bp of SSC. A total of 111 genes in *K. galanga* and 113 genes in *K. elegans* comprised 79 protein-coding genes and 4 ribosomal RNA (rRNA) genes, as well as 28 and 30 transfer RNA (tRNA) genes in *K. galanga* and *K. elegans*, respectively. The gene order, GC content and orientation of the two *Kaempferia* chloroplast genomes exhibited high similarity. The location and distribution of simple sequence repeats (SSRs) and long repeat sequences were determined. Eight highly variable regions between the two *Kaempferia* species were identified and 643 mutation events, including 536 single-nucleotide polymorphisms (SNPs) and 107 insertion/deletions (indels), were accurately located. Sequence divergences of the whole chloroplast genomes were calculated among related Zingiberaceae species. The phylogenetic analysis based on SNPs among eleven species strongly supported that *K. galanga* and *K. elegans* formed a cluster within Zingiberaceae. This study identified the unique characteristics of the entire *K. galanga* and *K. elegans* chloroplast genomes that contribute to our understanding of the chloroplast DNA evolution within Zingiberaceae species. It provides valuable information for phylogenetic analysis and species identification within genus *Kaempferia*.

## 1. Introduction

The genus *Kaempferia* belongs to the family Zingiberaceae, which consists of approximately 50 species in the world [1,2,3]. *Kaempferia* species are distributed in tropical Asia regions [1,2]. *Kaempferia* species are grown primarily for their ornamental foliage rather than for their flowers [3]. In addition, several species have long been cultivated for their medicinal properties [3]. *Kaempferia galanga* and *Kaempferia elegans* are valuable herbal medicine and ornamental plants in this genus, respectively. *K. galanga* is mainly distributed in the regions of Southern and Northwestern China (Guangdong, Guangxi, Guizhou, Yunnan and Sichuan provinces) and is widely cultivated in Southeast Asia; whereas *K. elegans* is only produced in Sichuan province of China and is commonly cultivated in tropical Asia regions [1,2]. Morphology data had been used to determine the differences between *K. galanga* and *K. elegans* [1,2,3]. From these studies, the two *Kaempferia* species had been characterized differently by leaf shape, petioles, flower color and rhizomes [1,2,3]. The leaves of *K. galanga* spread flat on ground, subsessile, green, orbicular, 7–20 × 3–17 cm, glabrous on both surfaces or villous abaxially, margin usually white, apex mucronate or acute; the leaves of *K. elegans* have petioles, to 10 cm, leaf adaxially green, abaxially pale green, oblong or elliptic, 13–15 × 5–8 cm, margin usually red, base rounded, apex acute. The petals are white in *K. galanga*, whereas the petals are purple in *K. elegans*. The rhizomes of *K. galanga* are pale green or greenish white inside, tuberous and fragrant; the rhizomes of *K. elegans* are bearing globose tubers with fibrous roots. *K. galanga* can be used as aromatic medicinal plant and also as ornamental plant with important economic value, whereas *K. elegans* is mainly used as potted plant with ornamental value. In detail, the rhizomes of *K. galanga* have been used as aromatic stomachic, with effects of dissipating cold, dampness, warm the spleen and stomach, also used as flavoring spices, for example, a famous Guangdong delicacy of sand ginger salted chicken [2]. With increased demand for rhizomes of *K. galanga*, researches related to the high rhizomes yield for its cultivation has been lacking, leading to having a high price in the market. However, sometimes morphological identification of *Kaempferia* species was difficult owing to the morphological similarity of vegetative parts among species and other genera in Zingiberaceae, such as *Boesenbergia*, *Cornukaempferia*, *Curcuma* and *Scaphochlamys* [3,4]. In addition, intraspecific variation caused more complicated problems in the morphological taxonomy of the genus *Kaempferia* [3,4]. Based only on morphological characteristics, we could not conclusively distinguish and identify the *Kaempferia* species and hybrids and other genera species in Zingiberaceae. Therefore, the morphological classification and relationships within *Kaempferia* species need further investigation together with more molecular analyses. Combined evidences from morphology characteristics and chloroplast DNA have proven useful and powerful in species identification and phylogenetic relationships analysis [4,5,6,7,8].

In angiosperm plants, chloroplasts play an important role in photosynthesis and the metabolism of starch, fatty acids, nitrogen, amino acids and internal redox signals [9,10,11]. In general, the chloroplast genomes of angiosperms encode 110–130 genes with a size range of 120-180 kb, which have a typical quadripartite structure consisting of a large single copy (LSC) region, a small single copy (SSC) region and two copies of inverted repeats (IRs) [12,13,14,15]. As the chloroplast is the center of photosynthesis, the research of the chloroplast genome is important to discover the mechanisms of plant photosynthesis. The third-generation sequencing platform PacBio utilizes a single-molecule real-time sequencing technology, which has been successfully used to determine many chloroplast genome sequences [16,17,18]. The main advantage of this method is the long read length of over 10 kb on average, which provides many benefits in genome assembly, including longer contigs and fewer unresolved gaps [16,17]. However, PacBio sequencing has high rates of random error in its single-pass reads; therefore, in combination with Illumina sequencing can reduce these random errors [17,18].

In this study, we sequenced and analyzed the complete chloroplast genomes from *K. galanga* and *K. elegans*, respectively, using Illumina and PacBio sequencing. We also characterized the long repeats and SSRs detected in the genome, including repeat types, distribution patterns and so on. Comparative sequence analysis and phylogenetic relationships were also analyzed among other members in the family Zingiberaceae. These results will improve the genetic information of the genus *Kaempferia* we already have and will be beneficial for DNA molecular studies in *Kaempferia*.

## 2. Results

### 2.1. Chloroplast Genome Organization of Two Kaempferia Species

The complete chloroplast genome of *K. galanga* and *K. elegans* consisted of a single circular molecule with quadripartite structure (Figure 1). The sizes of *K. galanga* and *K. elegans* chloroplast genomes were 163,811 and 163,555 bp, respectively. They consisted of a pair of inverted repeats (IRs) of 29,797 bp in *K. galanga* and 29,773 bp in *K. elegans*, a large single copy (LSC) region of 88,405 bp in *K. galanga* and 88,020 bp in *K. elegans* and a small single copy (SSC) region of 15,812 bp in *K. galanga* and 15,989 bp in *K. elegans* (Figure 1 and Table 1). The GC content of the genomes was 36.1% both in *K. galanga* and *K. elegans* but the IR regions had higher GC contents (41.2% and 41.1% in *K. galanga* and *K. elegans*, respectively) than that of the LSC regions (33.9% both in *K. galanga* and *K. elegans*) and SSC regions (29.5% and 29.4% in *K. galanga* and *K. elegans*, respectively). Approximately 48.3–50.7% of the two *Kaempferia* species chloroplast genomes consisted of protein-coding genes (83,172 bp in *K. galanga* and 79,117 bp in *K. elegans*), 1.7% of tRNAs (2870 bp *K. galanga* and 2852 bp in *K. elegans*) and 5.5% of rRNAs (9046 bp in *K. galanga* and 9046 bp in *K. elegans*). For protein-coding regions, the AT content for the first, second and third codons were 55.4%, 62.6% and 71.1% in *K. galanga*, respectively and 66.9%, 56.7% and 64.7% in *K. elegans*, respectively (Table 1). The non-coding regions consisting of introns, pseudogenes and intergenic spacers accounted for 49.3% and 51.7% for the *K. galanga* and *K. elegans* chloroplast genomes, respectively (Table 1).

There were 111 predicted genes in the *K. galanga* chloroplast genome including 79 protein-coding genes, 28 tRNA genes and 4 rRNA genes, while 113 genes predicted in the *K. elegans* chloroplast genome consisted of 79 protein-coding genes, 30 tRNA genes and 4 rRNA genes (Table 2). Among the protein-coding genes in *K. galanga* chloroplast genome, 61 genes were located in the LSC region, 12 genes were in the SSC region and 8 genes were duplicated in the IR regions (Appendix A). In total, there were 18 intron-containing genes in the *K. galanga* chloroplast genome, 16 of which contained one intron and two of which (*ycf3* and *clpP*) contained two introns (Table 3). Among the protein-coding genes in the *K. elegans* chloroplast genome, 63 genes were located in the LSC region, 12 genes were in the SSC region and 6 genes were duplicated in the IR regions (Appendix A). In total, there were 17 intron-containing genes in the *K. elegans* chloroplast genome, 15 of which contained one intron and two of which (*ycf3* and *clpP*) contained two introns (Table 3).

Relative synonymous codon usage (RSCU) is the ratio between frequency of use and expected frequency of a particular codon. RSCU values <1.00 indicate use of a codon less frequent than expected, while codons used more frequently than expected have a score of >1.00 [19,20]. The codon usage of the *K. galanga* and *K. elegans* chloroplast genomes are summarized in Appendix A. Protein-coding genes comprised 27,724 and 27,675 codons in both the *K. galanga* and *K. elegans*, respectively. Among these codons, those for leucine and isoleucine were the most common in both *K. galanga* and *K. elegans* chloroplast genomes (Figure 2 and Appendix A). The use of the codons ATG and TGG, encoding Met and Trp respectively, exhibited no bias (RSCU = 1.00) in these two *Kaempferia* species chloroplast genomes (Appendix A). Codons ending in A and/or U accounted for 71.1% and 64.7% of all protein-coding gene codons of the chloroplast genomes of *K. galanga* and *K. elegans*, respectively (Table 1 and Appendix A). The findings also revealed that all of the types of preferred synonymous codons (RSCU>1.00) ended with A or U except for *trL-CAA* in these two *Kaempferia* species (Appendix A).

### 2.2. Analysis of SSRs and Long Repeats

SSRs or microsatellites, are tandem repeat sequences consisting of 1-6 nucleotide repeat units and are widely distributed in chloroplast genomes [5,7,12]. SSRs were detected using MISA in both *Kaempferia* species chloroplast genomes. We detected 240 and 248 SSRs in *K. galanga* and *K. elegans* chloroplast genomes, respectively. Mononucleotide motifs were the most abundant type of repeat and dinucleotides were the second most abundant in both *Kaempferia* species chloroplast genomes (Figure 3 and Appendix A). There were 177 momo-, 32 di-, 6 tri-, 21 tetra-, 3 penta- and one hexa-nucleotide SSRs in *K. galanga* chloroplast genome; by contrast, there were 188 momo-, 33 di-, 7 tri-, 19 tetra- and 1 penta-nucleotide SSRs in *K. elegans* chloroplast genome (Figure 3). The majority of SSRs were located in the LSC regions rather than in IR and SSC regions of both *Kaempferia* species chloroplast genomes (Figure 3 and Appendix A). SSRs were more abundant in non-coding regions than in coding regions of both genomes (Figure 3). Furthermore, almost all SSR loci were composed of A or T, which contributed to the bias in base composition (A/T; both 63.9%) in the chloroplast genomes of the two *Kaempferia* species.

Long repeat sequences in the *K. galanga* and *K. elegans* chloroplast genomes were analyzed by REPuter and results shown in Figure 4 and Appendix A. In the *K. galanga* chloroplast genome, 21 forward repeats, 20 palindrome repeats, 5 reverse repeats and 4 complement repeats were detected (identity>90%) (Figure 4 and Appendix A). In comparison, in the *K. elegans* chloroplast genome, 26 forward repeats, 17 palindrome repeats, 4 reverse repeats and 2 complement repeats were detected (Figure 4 and Appendix A). Out of the 50 repeats in *K. galanga* chloroplast genome, 38 repeats (76.0%) were 30–39 bp long, 9 repeats (18.0%) were 40–49 bp long and 3 repeats (6.0%) were ≥50 bp long (Figure 4 and Appendix A). By contrast, of the 49 repeats in *K. elegans* chloroplast genome, 37 repeats (75.5%) were 30–39 bp long, 6 repeats (12.2%) were 40–49 bp long and 6 repeats (12.2%) were ≥50 bp long (Figure 4 and Appendix A). The majority of these repeats were mainly forward and palindromic types with lengths mainly in the range of 30–50 bp in both *Kaempferia* species.

### 2.3. IR Contraction and Expansion

A detailed comparison was performed for four junctions, LSC/IRa, LSC/IRb, SSC/IRa and SSC/IRb, between the two IRs (IRa and IRb) and the two single-copy regions (LSC and SSC) among *A. zerumbet*, *C. flaviflora* and *Z. spectabile* in comparison to *K. galanga* and *K. elegans* (Figure 5). Although the IR region of the five Zingiberaceae species chloroplast genomes was highly conserved, structure variation was still found in the IR/SC boundary regions. As shown in Figure 5, the *rpl22*-*rps19* genes were located in the junctions of the LSC/IRb regions in *K. galanga*, *K. elegans*, *A. zerumbet* and *C. flaviflora*, though the *trnM*-*ycf2* sequence in *Z. spectabile*, one of which was missing the *rpl22*/-*rps19* gene in the junctions of the LSC/IRb regions. The *ycf1*-*ndhF* genes were located at the junctions of the IRb/SSC regions in the five Zingiberaceae species chloroplast genomes. The *ndhF* gene was 23, 98, 251, 133 and 33 bp from the IRb/SSC border in *K. galanga*, *K. elegans*, *A. zerumbet*, *C. flaviflora* and *Z. spectabile*, respectively (Figure 5). The SSC/IRa junctions in the five Zingiberaceae species chloroplast genomes were crossed by the *ycf1* gene, with 665–3888 bp in the IRa region. Like the IRb/SSC boundary regions, the IRa/LSC regions were also variable. The *rps19*-*psbA* genes were located in the junctions of the IRa/LSC regions in *K. galanga*, *K. elegans*, *A. zerumbet* and *C. flaviflora*, though the *trnH*-*psbA* genes in *Z. spectabile*, one of which was missing the *rps19* gene in the junctions of the IRa/LSC region. The *rps19*-*psbA* genes of *K. elegans* were located at the junctions of IRa/LSC regions with 136 and 123 bp, respectively, separating the spacer from the end of the IRa region. However, in *Z. spectabile*, the *trnH* gene was the last gene at one end of the IRa region, 256 bp away from the IRa/LSC border. Overall, contraction and expansion of the IR regions was detected across the five Zingiberaceae species chloroplast genomes.

### 2.4. Comparative Chloroplast Genomic Analysis

To characterize genome divergence, we performed multiple sequence alignments between the five Zingiberaceae species chloroplast genomes using the program mVISTA, with *K. galanga* being used as a reference (Figure 6). The comparison demonstrated that the two IR regions were less divergent than the LSC and SSC. Moreover, the coding regions are more conserved than the non-coding regions. The most highly divergent regions among the five chloroplast genomes were found among the intergenic spacers, including *trnH*-*psbA*, *rps16*-*psbK*, *atpH*-*atpI*, *petN*-*psbM*, *trnE*-*psbD psbC*-*rps14*, *rps4*-*ycf3*, *rps4*-*ndhJ*, *ndhC*-*atpE*, *ycf4*-*cemA* and *petA*-*psbJ* in LSC as well as *rpl32*-*ccsA*, *psaC*-*ndhG* and *ndhG*-*ndhI* in SSC. Higher divergence in the coding regions was found in the *matK*, *rpoA*, *rps16*, *rps19*, *ndhF*, *ccsA*, *psaC*, *ndhD*, *ndhE*, *ndhG*, *ndhI*, *ndhA* and *ycf1* sequences.

Furthermore, sliding window analysis using DnaSP detected highly variable regions in the chloroplast genomes between *K. galanga* and *K. elegans* (Figure 7A). The average value of nucleotide diversity (Pi) was 0.01075. The IR regions showed lower variability than the LSC and SSC regions. There were 7 mutational hotspots that exhibited remarkably higher Pi values (>0.03) and were located at the LSC and SSC regions, which included *trnS*-*trnG*, *rps12-clpP*, *psbT-psbN*, *ycf1-ndhF*, *ndhF-rpl32*, *psaC-ndhE* and *ccsA-ndhD* regions from the chloroplast genomes (Figure 7A). By contrast, there was only 1 mutational hotspot (*rpl2-trnH*) that exhibited remarkably higher Pi values (>0.03) located at the IR regions (Figure 7A).

Figure 7B showed that the average value of Pi was 0.01591 among two *Kaempferia* species, *A. zerumbet*, *C. flaviflora* and *Z. spectabile*. The Pi values of these five species were commonly higher than those of the two *Kaempferia* species. Particularly, seven highly divergent loci showed remarkably higher Pi values (>0.045), including *trnS*-*trnG*, *psbT-psbN*, *trnH-rpl2*, *trnI-ycf2*, *ccsA-ndhD*, *psaC-ndhE* and *ycf2-trnI* regions from the chloroplast genomes (Figure 7B). These regions may be undergoing rapid nucleotide substitution at the species level, indicating potential application of molecular markers for plant identification and phylogenetic analysis.

The chloroplast genomes of *K. galanga* and *K. elegans* were found to show a 256 bp difference in length (Table 1). In addition to the total length difference, we assessed SNP and Indel variations between the two *Kaempferia* species chloroplast genomes in their entirety. There were 536 SNPs identified in the two chloroplast genomes (Appendix A). The most frequently occurring mutations were located in intergenic region, which included 357 SNPs. The coding regions contained 91 synonymous SNPs, 87 nonsynonymous SNPs and 1 stop mutation. There were 107 indels in the chloroplast genome identified between *K. galanga* and *K. elegans* (Appendix A), including 47 deletions and 60 insertions. Of the 107 indel markers between *K. galanga* and *K. elegans* genomes, the longest indels (10 bp) were located within the two intergenic sequences (*petA*-*psbJ* and *atpH*-*atpI*) and two coding sequences (*atpF* and *rps12*).

### 2.5. Phylogenetic Analysis

In this study, phylogenetic trees were constructed with SNPs from eleven species using ML and MP methods, respectively, including nine Zingiberaceae plants and using *C. pulverulentus* and *C. indica* as outgroups (Figure 8). Both the ML and MP phylogenetic trees strongly indicated that *K. galanga* and *K. elegans* formed a cluster within Zingiberaceae and the *C. pulverulentus* and *C. indica* species were clearly separated from Zingiberaceae species (Figure 8). Among nine Zingiberaceae species, they were clustered into four clusters. The first cluster comprised the genus *Kaempferia* (*K. galanga* and *K. elegans*). The second cluster comprised the two genera—*Zingiber* and *Curcuma* (*Z. spectabile*, *C. flaviflora* and *C. roscoeana*). The third cluster comprised the genus *Amomum* (*A. kravanh* and *A. compactum*). The fourth cluster comprised the genus *Alpinia* (*A. zerumbet* and *A. oxyphylla*).

### 2.6. Potential RNA Editing Sites

In the present study, potential RNA editing sites were predicted for 34 genes; as a result, a total of 54 and 80 RNA editing sites were identified in the *K. galanga* and *K. elegans* chloroplast genomes, respectively (Appendix A). No potential editing sites were identified in seven genes (*petG*, *petL*, *psaB*, *psaI*, *psbL*, *rpl2*, *rpl23*) in both chloroplast genomes. Of the 54 editing sites, which occurred in 21 genes, 15 (27.8%) and 39 (72.2%) were located at the first and the second codon position, respectively, in *K. galanga*. Of the 80 editing sites, which occurred in 26 genes, 21 (26.2%) and 59 (73.8%) were located at the first codon and the second codon position, respectively, in *K. elegans*. No editing sites were found at the third codon position in both *Kaempferia* species.

We also observed that RNA editing sites were all C to U conversion both in *K. galanga* and *K. elegans*. In *K. galanga*, the *ndhB* gene was predicted to have ten editing sites; *accD* and *matK*, five; *rpoB* and *rpoC2*, four; *rpl20*, *rpoA* and *rps14*, three; *atpB*, *petB*, *psbE* and *rpoC1*, two; and one each in *atpA*, *atpF*, *atpI*, *ccsA*, *clpP*, *psbB*, *psbF*, *rps2* and *rps8*. Meanwhile in *K. elegans*, the gene of *ndhB* was predicted to have ten editing sites; *ndhA* and *ndhD*, seven; *accD*, *matK*, *ndhF* and *rpoC2*, five; *rpoB* and *ycf3*, four; *rpl20*, *rpoA* and *rps14*, three; *atpB*, *ndhG*, *petB*, *psbB* and *rpoC1*, two; and one each in *atpA*, *atpF*, *atpI*, *ccsA*, *clpP*, *psbF*, *rps2*, *rps8* and *rps16*.

## 3. Materials and Methods

### 3.1. Plant Material and DNA Isolation

Fresh leaves were collected from potted *K. galanga* and *K. elegans* plants, respectively, from greenhouse in environmental horticulture research institute, Guangdong academy of agricultural sciences, Guangzhou, China. Total chloroplast DNA was extracted from about 100 g of leaves using the sucrose gradient centrifugation method as improved by Li et al. [21]. The chloroplast DNA concentration for each sample was estimated using an ND-2000 spectrometer (Nanodrop technologies, Wilmington, DE, USA), whereas visual examination was performed using gel electrophoresis.

### 3.2. Chloroplast Genome Sequencing and Genome Assembly

The chloroplast DNA was first fragmented into 300–500 bp using a Covaris M220 Focused-ultrasonicator (Covaris, Woburn, MA, USA) and used to construct short-insert libraries (insert size about 430 bp) according to the manufacturer’s instructions (Illumina, San Diego, CA, USA). The short fragments were sequenced using an Illumina Hiseq X Ten platform (Novogene, Beijing, China). The Illumina raw reads were cleaned by removing the adapter sequences and low quality sequences, which included the reads with ambiguous nucleotides and ones containing more than 10% nucleotides in read with Q-value ≤ 20 and short reads (length < 50 bp).

The chloroplast DNA was also fragmented into 8-10 kb fragments, which were subjected to DNA sequencing following the standard protocol provided by PacBio platform (Novogene, Beijing, China). The PacBio raw reads were pre-processed by trimming the adapter sequences, low quality (Q < 0.80) reads, short reads (length < 100 bp) and short subreads (length <500 bp).

Initially, the Illumina clean reads were assembled using SOAPdenovo (version 2.04, Hongkong, China) with default parameters into principal contigs [22] and all contigs were sorted and joined into a single draft sequence using the software Geneious version 11.0.4 (Auckland, New Zealand) [23]. Next, BLASR software (San Diego, CA, USA) was used to compare the PacBio clean data with the single draft sequence and to extract the correction and error correction [24]. Next, the corrected PacBio clean data were assembled using Celera Assembler (version 8.0, Rockville, MD, USA) with default parameters, thus generating scaffolds [25]. Next, the assembled scaffolds were mapped back to the Illumina clean reads using GapCloser (version 1.12, Hongkong, China) for gap closing [22]. Finally, the redundant fragments sequences were removed, thus generating the final assembled chloroplast genomic sequence.

### 3.3. Chloroplast Genome Annotation and Codon Usage

The initial gene annotation of the chloroplast genome was carried out with BLAST homology searches and DOGMA (Dual Organellar Genome Annotator) [26]. tRNA genes were identified using tRNAscanSE with default settings [27]. The gene homologies were confirmed by comparing them with National Center for Biotechnology Information (NCBI)’s non-redundant (Nr) protein database, Clusters of orthologous groups (COG) for eukaryotic complete genomes database (http://www.ncbi.nlm.nih.gov/COG), Kyoto Encyclopedia of Genes and Genomes (KEGG) (http://www.kegg.jp/), Gene Ontology (GO) (http://www.geneontology.org) and SWISS-PROT (http://web.expasy.org/docs/swissprotguideline.html) databases. The structural features of chloroplast genome maps were drawn using OGDRAWv1.2 (Potsdam-Golm, Germany) [28]. Codon usage was determined for all protein-coding genes. To examine the deviation in synonymous codon usage, the relative synonymous codon usage (RSCU) was calculated using MEGA6 software (Version 6.0, Jeddah, Saudi Arabia) [29]. Amino acid (AA) frequency was also calculated and expressed by the percentage of the codons encoding the same amino acid divided by the total number of codons. The final chloroplast genomic sequences have been submitted to GenBank under accession numbers MK209001 and MK209002 for *K. galanga* and *K. elegans*, respectively.

### 3.4. SSRs and Long Repeat Structure

SSRs were identified using MIcroSAtellite (MISA) [30]. The parameters for SSRs were adjusted for identification of perfect mono-, di-, tri-, tetra-, pena- and hexanucleotide motifs with a minimum of 8, 5, 4, 3, 3 and 3 repeats, respectively. The online REPuter software was used to identify and locate forward, palindrome, reverse and complement repeat sequences with repeat sizes ≥30 bp and sequences identity ≥90% [31].

### 3.5. Comparative and Divergence Analysis of Chloroplast Genomes of K. galanga and K. elegans

The complete chloroplast genome of *K. galanga* was employed as a reference and was compared with the chloroplast genomes of *K. elegans*, *Alpinia zerumbet* (JX088668), *Curcuma flaviflora* (KR967361) and *Zingiber spectabile* (JX088661), the last three of which were obtained from GenBank, using mVISTA program (http://genome.lbl.gov/vista/mvista/about.shtml) in the Shuffle-LAGAN mode [32]. To calculate nucleotide variability (Pi) between *K. galanga* and *K. elegans* chloroplast genomes, sliding window analysis was performed using DnaSP version 5.1 software [33] with window length of 600 bp and the step size of 200 bp.

The complete chloroplast sequences of *K. galanga* and *K. elegans* were also aligned using MUMmer software (Maryland, USA) [34] and adjusted manually where necessary using Se-Al 2.0 [35]. The single nucleotide polymorphisms (SNPs) and insertion/deletions (indels) were recorded separately as well as their locations in the chloroplast genome.

### 3.6. Phylogenetic Analysis

A molecular phylogenetic tree was constructed using SNP arrays from 11 species including *K. galanga* and *K. elegans*. Among these 11 species, nine complete chloroplast genome sequences were downloaded from NCBI: *A. zerumbet* (JX088668), *C. flaviflora* (KR967361), *Z. spectabile* (JX088661), *C. roscoeana* (NC_022928.1), *Alpinia oxyphylla* (NC_035895.1), *Amomum kravanh* (NC_036935.1), *Amomum compactum* (NC_036992.1), *Costus pulverulentus* (KF601573) and *Canna indica* (KF601570). *K. galanga* chloroplast genome was used as reference. *Costus pulverulentus* and *Canna indica* were set as outgroups of the family Zingiberaceae. Firstly, using MUMmer software [34], each chloroplast genome above was compared globally with the reference genome and the difference between each chloroplast genome and the reference genome found and preliminary filtering performed to detect the potential SNP sites. Secondly, the sequence of 100 bp on each side of the reference sequence SNP site was extracted and the extracted sequence and assembly results were compared using the BLAT software [36,37] to verify the SNP site. If the length of the alignment is less than 101 bp, it is considered to be a non-trusted SNP and is removed; if compared several times, the SNP that is considered to be a duplicate region and will also be removed; and finally a reliable SNP will be obtained. Thirdly, for each chloroplast genome, all SNPs are connected in the same order to obtain a sequence in FASTA format. Multiple FASTA format sequences alignments were carried out using ClustalX version 1.81 [38]. To examine the phylogenetic applications of rapidly evolving SNP markers, the maximum likelihood (ML) and maximum parsimony (MP) methods with 1000 bootstrap replicates were employed to construct phylogenetic trees using MEGA6 software, respectively [29].

### 3.7. RNA Editing Analysis

Thirty-four protein-coding genes of *K. galanga* and *K. elegans* chloroplast genomes were used to predict potential RNA editing sites using the online program Predictive RNA Editor for Plants (PREP) suite (http://prep.unl.edu/) with a cutoff value of 0.8 (Bielefeld, Germany) [39].

## 4. Discussion

In this study, we obtained the complete chloroplast genomes of *K. galanga* and *K. elegans* by using Illumina and PacBio sequencing, ranging from 163.5-163.8 kb in length. Both chloroplast genomes exhibit a typical quadripartite structure, as reported for other Zingiberaceae species, such as *A. oxyphylla*, *A. zerumbet*, *C. flaviflora*, *Z. spectabile*, *C. roscoeana*, *A. compactum* and *A. kravanh* [12]. Both genomes encode about 111-113 genes, including 79 protein-coding genes, 4 rRNA genes as well as 28 and 30 tRNA genes distributed throughout their genomes, respectively. This conformed with the protein-coding genes found in other Zingiberaceae members [12].

The molecular markers obtained from chloroplast genome sequences such as highly variable sequences, SSRs, SNPs and indels are useful tools in research. In *Camellia* species, 1.5% high divergent sequences were used for phylogenetics, taxonomy and species identification [40]. In this study, eight highly variable regions had been detected between *K. galanga* and *K. elegans* chloroplast genomes, including *trnS*-*trnG*, *rps12-clpP*, *psbT-psbN*, *ycf1-ndhF*, *ndhF-rpl32*, *psaC-ndhE*, *ccsA-ndhD* and *rpl2-trnH* (Figure 7A). As our results displayed, most of them occurred in the LSC and SSC regions but only one in the IR regions. Among these highly variable regions, *ndhF-rpl32*, *ccsA-ndhD*, *trnS*-*trnG*, *psbT-psbN* and *psaC-ndhE* had been reported as highly variable regions in many species, such as *Papaver* [5], *Machilus* [41], *Citrullus* [42], *Fagopyrum* [43], *Citrus* [44] and *Oryza* [45]. In addition, the *ndhF-rpl32*, *ccsA-ndhD* and *trnS*-*trnG* regions had been used as molecular markers for phylogenetic analysis, to resolve origin problems and phylogeographic studies [13,41,42,43,45]. Therefore, these eight highly variable regions could serve to enrich the molecular marker resources of genus *Kaempferia* in studies of phylogeny, evolution and species identification.

Besides the highly variable regions, we were able to retrieve SSRs and long repeats. Of the 240 SSRs identified in *K. galanga*, 64.58% (155 SSRs) were located in the LSC region, 16.66% (49 SSRs) in the SSC region and 18.75% (45 SSRs) in the IR regions. In contrast, out of the 248 SSRs identified in *K. elegans*, 64.11% (159 SSRs) were present in the LSC region, 16.93% (42 SSRs) in the SSC region and the remaining 19.75% (49 SSRs) located in the IR regions, as reported in other plants like *A. kravanh* [12], *Talinum paniculatum* [20] and *Oryza minuta* [45]. Among the SSRs types, the most abundant was found to be mononucleotides in both *K. galanga* and *K. elegans* (Figure 3). These findings were in agreement with results from previous studies in *A. kravanh* [12], *T. paniculatum* [20] and buckwheat species [43] but were different from *O. minuta* which possessed a majority of dinucleotide repeat motif SSRs [45]. AT/AT (12.5%) was the most frequent dinucleotide motifs followed by AAAT/ATTT in both *K. galanga* and *K. elegans*, respectively (Figure 3). These SSRs and long repeats identified in our study could be useful in molecular studies, such as genetic diversity and phylogenetic relationship analysis, species identification and evolution studies [7,12,21,40,43].

In the present study, we also identified 536 SNPs and 107 indels between the two *Kaempferia* species (Appendix A). From the SNPs results, 536 nucleotide substitutions were detected between *K. galanga* and *K. elegans* chloroplast genomes. It indicated that the nucleotide substitution events in the chloroplast genomes of *Kaempferia* species were more than that between species of *Oryza*, *Machilus*, cultivated *Fagopyrum*, *Citrus and Panax but* less than species of *Solanum* and wild *Fagopyrum*. Comparative analysis of chloroplast genomes found 159 SNPs between *Oryza nivara* and *O. sativa* [46], 231 SNPs between *Machilus yunnanensis* and *M. balansae* [41], 317 SNPs between cultivated species *Fagopyrum dibotrys* and *F. tataricum* [43], 330 SNPs between *Citrus sinensis* and *C. aurantiifolia* [44], 464 SNPs between *Panax notoginseng* and P. ginseng [47], 591 SNPs between *Solanum tuberosum* and *S. bulbocastanum* [48], 6260 SNPs between wild species *F. luojishanense* and *F. esculentum* [43]. Out of the 107 indels found between *K. galanga* and *K. elegans* chloroplast genomes, two longest intergenic sequences *petA*-*psbJ* and *atpH*-*atpI* were detected. The *petA*-*psbJ* and partial *psbA*-*trnH* spacer sequences can be used for species identification of most *Kaempferia* and outgroup species [4]. Similarly, a single large 241-bp deletion in *S. tuberosum* clearly discriminated a cultivated potato from the wild potato species *S. bulbocastanum* [48]. The indels and SNPs of 12 Triticeae species chloroplasts were used to estimate wheat, barley, rye and their relatives evolution [49]. These indels and SNPs found in our study could be useful in phylogenetic analysis, species identification and evolutionary studies as well as the 65 indels detected between *M. yunnanensis* and *M. balansae* [41], 156 indels between *P. notoginseng* and P. ginseng [47] and 53 indels between *Aconitum pseudolaeve* and *A. longecassidatum* [50].

The chloroplast genome sequences provide a useful genomic resource for phylogenetic studies and many studies have successfully used protein-coding sequences or whole chloroplast genome sequences in these analyses [7,12,20,43]. Specifically, the chloroplast *psbA*-*trnH* and partial *petA*-*psbJ* sequences and *matK* gene had been utilized in Zingiberaceae species phylogenetic studies before [4,51]. In this study, we constructed phylogenetic trees using ML and MP methods based on SNPs commonly present in the chloroplast genomes of eleven species, including two *Kaempferia* species from the current study. Our phylogenetic analysis clearly revealed that the two *Kaempferia* species clustered together, with bootstrap values of 100%, as well as *Amomum* and *Alpinia* species, which segregated in two sister clades (Figure 8). In a previous study, a phylogenetic tree constructed by using whole chloroplast genome sequences strongly supported the position in the Zingiberaceae of *A. kravanh* as a sister of the closely related species *A. zerumbet* [12]. Our phylogenetic trees using SNPs were in broad agreement with the previous study [12]. Therefore, phylogenetic analysis using SNPs among chloroplast genome sequences could provide useful information for revealing relationships among Zingiberaceae species.

In conclusion, we assembled and analyzed the complete chloroplast genomes of *K. galanga* and *K. elegans* and compared them with other Zingiberaceae species for the first time. The chloroplast genomes organization, gene order, GC content and codon usage of the two *Kaempferia* species showed high similarity. The location and distribution of SSRs and long repeat sequences were determined. Eight highly variable regions between the two *Kaempferia* species were identified and 643 mutation events, including 536 SNPs and 107 indels, were accurately located. Sequence divergences of chloroplast genomes were also calculated for the two *Kaempferia* species and related Zingiberaceae species. The phylogenetic analysis based on SNPs among eleven species strongly supported that *K. galanga* and *K. elegans* formed a cluster within Zingiberaceae. Our results provided insights into the characteristics of the entire *K. galanga* and *K. elegans* chloroplast genomes and the phylogenetic relationships within Zingiberaceae species.

## Figures and Tables

**Figure 1 molecules-24-00474-f001:**
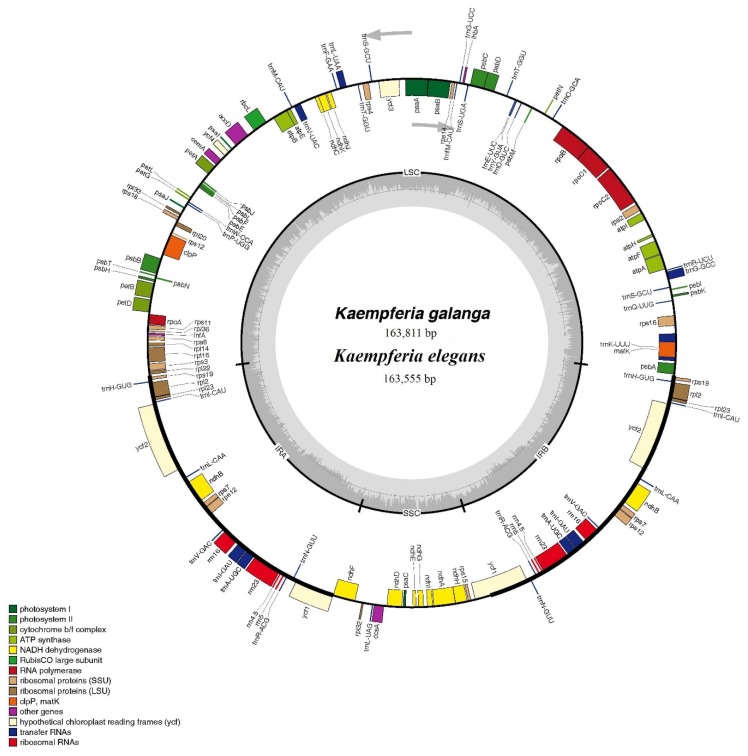
Circular gene map of chloroplast genomes of two *Kaempferia* species. The gray arrowheads indicate the direction of the genes. Genes shown inside the circle are transcribed clockwise and those outside are transcribed counterclockwise. Different genes are color coded. The innermost darker gray corresponds to GC content, whereas the lighter gray corresponds to AT content. IR, inverted repeat; LSC, large single copy region; SSC, small single copy region.

**Figure 2 molecules-24-00474-f002:**
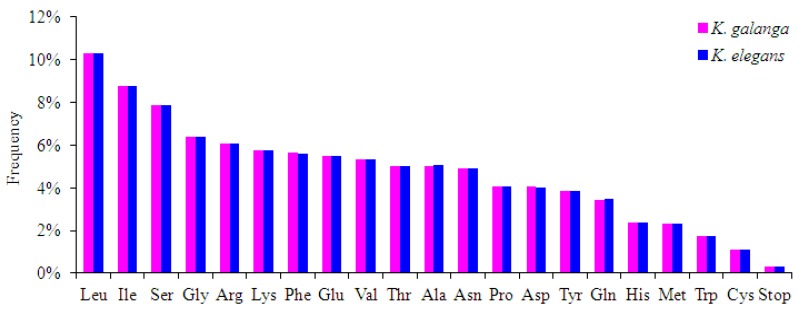
Amino acid frequencies in *K. galanga* and *K. elegans* protein-coding sequences.

**Figure 3 molecules-24-00474-f003:**
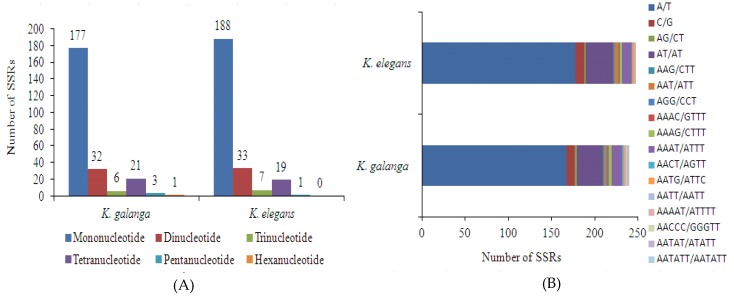
Distribution of SSRs in the chloroplast genomes of *K. galanga* and *K. elegans*. (**A**) Number of different SSR types detected in the two *Kaempferia* species chloroplast genomes; (**B**) Frequency of identified SSR motifs in different repeat class types; (**C**) SSR distribution in different genomic regions of two *Kaempferia* species chloroplast genomes; (**D**) SSR distribution between coding and non-coding regions of two *Kaempferia* species chloroplast genomes.

**Figure 4 molecules-24-00474-f004:**
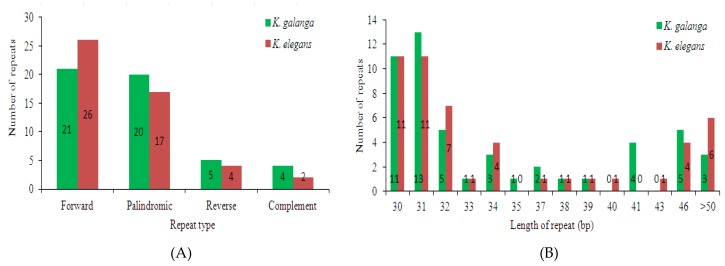
Analysis of long repeat sequences in the chloroplast genomes of *K. galanga* and *K. elegans*. (**A**) Frequency of long repeats types; (**B**) Frequency of long repeats by length.

**Figure 5 molecules-24-00474-f005:**
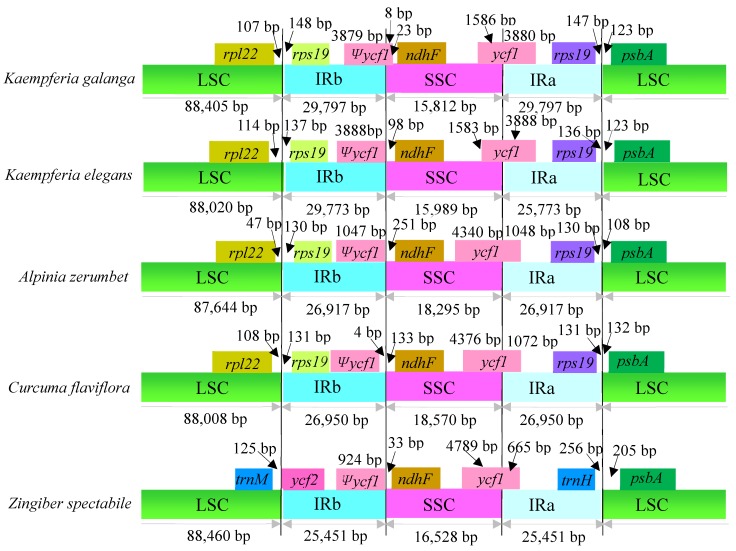
Comparison of the borders of the LSC, SSC and IR regions among five Zingiberaceae chloroplast genomes. Ψ, pseudogenes. Boxes above the main line indicate the adjacent border genes. The figure is not to scale with respect to sequence length and only shows relative changes at or near the IR/SC borders.

**Figure 6 molecules-24-00474-f006:**
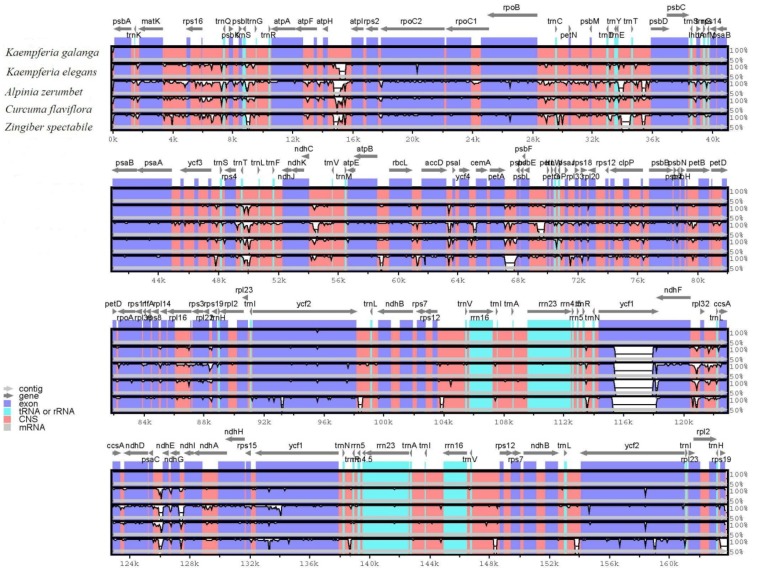
Comparison of five chloroplast genomes, with *K. galanga* as a reference using mVISTA alignment program. Gray arrows and thick black lines above the alignment indicate gene orientation. Purple bars represent exons, sky-blue bars represent transfer RNA (tRNA) and ribosomal RNA (rRNA), red bars represent non-coding sequences (CNS) and white peaks represent differences of chloroplast genomes. The *y*-axis represents the identity percentage ranging from 50 to 100%.

**Figure 7 molecules-24-00474-f007:**
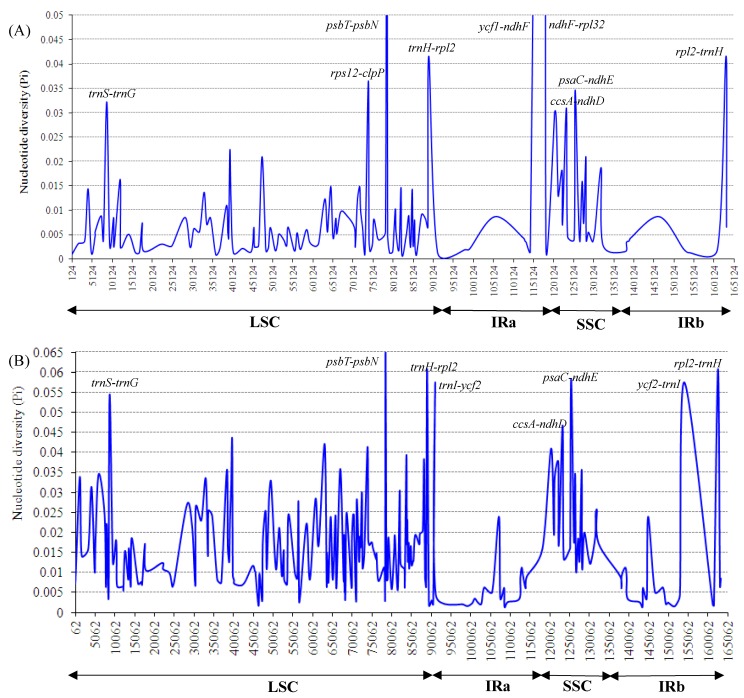
Sliding window analysis of the whole chloroplast genomes. Window length: 800 bp; step size: 200 bp. *X*-axis:position of the midpoint of a window. *Y*-axis: nucleotide diversity of each window. (**A**) Pi between *K. galanga* and *K. elegans*. (**B**) Pi among two *Kaempferia* species, *Alpinia zerumbet*, *Curcuma flaviflora* and *Zingiber spectabile*.

**Figure 8 molecules-24-00474-f008:**
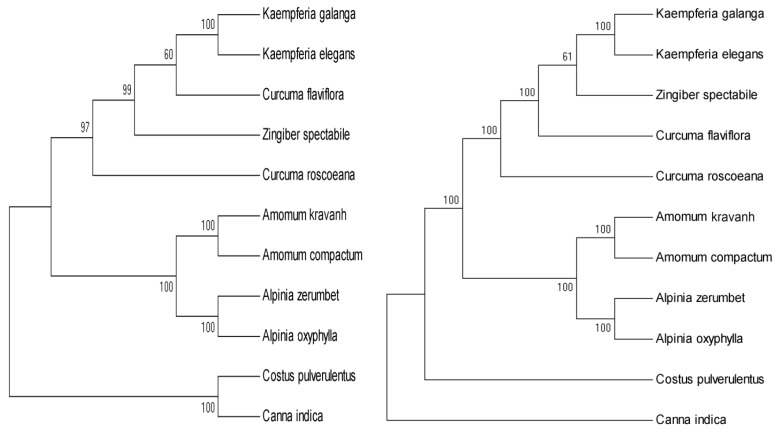
Phylogenetic trees constructed with SNPs from 11 species using maximum likelihood (ML, left) and maximum parsimony (MP, right) methods. Numbers at nodes on the tree indicate bootstrap values (>50%).

**Table 1 molecules-24-00474-t001:** Features of the chloroplast genomes of *K. galanga* and *K. elegans*.

Species	Regions	Positions	Length (bp)	T/U (%)	C (%)	A (%)	G (%)	AT/U (%)
*K. galanga*	Genome		163,811	32.2	18.3	31.7	17.7	63.9
	LSC		88,405	33.7	17.3	32.4	16.4	66.1
	IRa		29,797	28.8	19.8	30.0	21.2	58.8
	SSC		15,812	34.5	15.5	35.9	13.9	70.5
	IRb		29,797	28.8	19.8	30.0	21.2	58.8
	Protein coding genes		83,172	31.5	17.2	31.4	19.7	63.0
		1st position	27,724	23.9	18.2	31.4	26.3	55.4
		2nd position	27,724	32.4	20.0	30.1	17.3	62.6
		3rd position	27,724	38.3	13.3	32.8	15.5	71.1
	tRNA		2,870	24.9	23.6	22.0	29.3	47.0
	rRNA		9,046	18.6	23.6	26.1	31.5	44.8
*K. elegans*	Genome		163,555	32.2	18.3	31.7	17.7	63.9
	LSC		88,020	33.7	17.4	32.4	16.5	66.1
	IRa		29,773	28.8	19.8	30.1	21.3	58.9
	SSC		15,989	34.6	15.5	36.1	13.8	70.6
	IRb		29,773	28.8	19.8	30.1	21.3	58.9
	Protein coding genes		79,117	31.6	17.3	31.2	19.9	62.8
		1st position	26,372	35.0	14.3	31.9	18.8	66.9
		2nd position	26,372	26.6	19.4	30.1	23.9	56.7
		3rd position	26,372	33.3	18.2	31.4	17.1	64.7
	tRNA		2,852	24.9	23.7	22.0	29.4	46.9
	rRNA		9,046	18.7	23.6	26.1	31.5	44.8

**Table 2 molecules-24-00474-t002:** Genes present in the chloroplast genomes of *K. galanga* and *K. elegans*.

Category	Gene Name
Photosystem I	*psaA*, *psaB*, *psaC*, *psaI*, *psaJ*
Photosystem II	*psbA*, *psbB*, *psbC*, *psbD*, *psbE*, *psbF*, *psbH*, *psbI*, *psbJ*, *psbK*, *psbL*, *psbM*, *psbN*, *psbT*, *lhbA*
Cytochrome b/f	*petA*, *petB* *, *petD* *, *petG*, *petL*, *petN*
ATP synthase	*atpA*, *atpB*, *atpE*, *atpF* *, *atpH*, *atpI*
NADH dehydrogenase	*ndhA* *, *ndhB*(×2) *, *ndhC*, *ndhD*, *ndhE*, *ndhF*, *ndhG*, *ndhH*, *ndhI*, *ndhJ*, *ndhK*
Rubisco	*rbcL*
RNA polymerase	*rpoA*, *rpoB*, *rpoC1* *, *rpoC2*
Large subunit ribosomal proteins	*rpl2*(×2) *, *rpl14*, *rpl16* *, *rpl20*, *rpl22*, *rpl23*(×2), *rpl32*, *rpl33*, *rpl36*
Small subunit ribosomal proteins	*rps2*, *rps3*, *rps4*, *rps7*(×2), *rps8*, *rps11*, *rps12*(×2) *, *rps14*, *rps15*, *rps16* *, *rps18*, *rps19*(×2)
Other proteins	*accD*, *ccsA*, *cemA*, *clpP* *, *infA*, *matK*
Proteins of unknown function	*ycf1*(×2), *ycf2*(×2), *ycf3**, *ycf4*
Ribosomal RNAs	*rrn4.5*(×2), *rrn5*(×2), *rrn16*(×2), *rrn23*(×2)
Transfer RNAs	*trnA-UGC* (×2) *, *trnC-GCA*, *trnD-GUC*, *trnE-UUC*, *trnF-GAA*, *trnfM-CAU*, *trnG-GCC* (Kg×2, Ke) **, *trnG-UCC*, *trnH-GUG* (×2), *trnI-CAU* (×2), *trnI-GAU* (×2) *, *trnK-UUU* (×2) *, *trnL-CAA* (×2), *trnL-UAA* (×2) *, *trnL-UAG*, *trnM-CAU*, *trnN-GUU* (×2), *trnP-UGG*, *trnQ-UUG*, *trnR-ACG* (×2), *trnR-UCU*, *trnS-GCU* (Kg×2, Ke), *trnS-UGA*, *trnT-GGU* (Kg×2, Ke), *trnV-GAC* (×2), *trnV-UAC* (×2) *, *trnW-CCA*, *trnY-GUA*, *trnS-GGA* (Ke), *trnT-UGU* (Ke)

Kg: *K. galanga*; Ke: *K. elegans*; ×2: Gene with two copies; *: Genes containing introns both in *K. galanga* and *K. elegans*; **: Genes containing introns only in *K. galanga*.

**Table 3 molecules-24-00474-t003:** Genes with introns in the chloroplast genomes of *K. galanga* and *K. elegans*, including the exon and intron lengths.

Species	Gene	Location	Exon I (bp)	Intron I (bp)	Exon II (bp)	Intron II (bp)	Exon III (bp)
*K. galanga*	*trnA-UGC*	IR	38	801	35		
	*trnG-GCC*	LSC	14	711	48		
	*trnI-GAU*	IR	42	935	35		
	*trnK-UUU*	LSC	35	2646	37		
	*trnL-UAA*	LSC	35	536	50		
	*trnV-UAC*	LSC	37	598	38		
	*rps12* *	LSC/IR	114	-	231	540	27
	*rps16*	LSC	212	749	40		
	*rpl2*	IR	443	650	315		
	*rpl16*	LSC	402	1058	9		
	*petB*	LSC	6	783	642		
	*petD*	LSC	8	740	481		
	*atpF*	LSC	425	816	145		
	*ndhA*	SSC	518	1083	562		
	*ndhB*	IR	778	673	782		
	*rpoC1*	LSC	1632	726	432		
	*clpP*	LSC	252	636	306	856	60
	*ycf3*	LSC	153	794	228	714	132
*K. elegans*	*trnA-UGC*	IR	38	801	35		
	*trnI-GAU*	IR	42	935	35		
	*trnK-UUU*	LSC	35	2663	37		
	*trnL-UAA*	LSC	35	535	50		
	*trnV-UAC*	LSC	37	598	38		
	*rps12* *	LSC/IR	114	-	231	540	27
	*rps16*	LSC	212	729	40		
	*rpl2*	IR	432	659	387		
	*rpl16*	LSC	402	1056	9		
	*petB*	LSC	6	784	642		
	*petD*	LSC	8	741	481		
	*atpF*	LSC	411	816	144		
	*ndhA*	SSC	540	1079	552		
	*ndhB*	IR	756	700	777		
	*rpoC1*	LSC	1632	728	432		
	*clpP*	LSC	255	636	291	854	69
	*ycf3*	LSC	153	794	228	723	132

* The *rps12* gene is divided into 5′-*rps12* in the LSC region and 3′-*rps12* in the IR region.

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
