# Peer review of "Complete Chloroplast Genome Sequences of Kaempferia Galanga and Kaempferia Elegans: Molecular Structures and Comparative Analysis"

_molecules, 2019, doi:10.3390/molecules24030474_

Round 1

Reviewer 1 Report

The manuscript by Li and colleagues describes the determination of two chloroplast (cp) genomes from two Kaemferia species. The authors perform detailed (although fairly standard) comparative analyses of the two plus closely related cp genomes.

Specific Points (mainly language considerations)

1. L41 commonly cultivated

2. L42 [1, 2]. Morphology data …..

3. L43 [1-3]. From these studies, the two …

4. L45—this long sentence needs additional verbs…are subsessile, green…

5. L50 K. elegans bear globose ….

6. L50 Should this be aromatic? Rather than romatic?

7. L53 dissipating cold..

8. L55 research related …has been lacking leading to…having a high prce…

9. L60. Based only on…conclusively distinguish

10. L64 need further investigation together with more molecular analyses.

11. L64 DNA have proven useful and powerful in….. relationship analyses [4-8].

12. L71 As the chloroplast…

13. L74 used to determine many chloroplast genome sequences [16-18].

14. L82 The reader will not know what the authors are thinking of when the authors use “etc”

15. L89 respectively, from greenhouses in environmental…

16. L96 was first fragmented

17. L100 adapter sequences and low quality sequences, which…

18. L103 and L378. Use either kb or kbp—both are used in the ms

19. L105-L110. The details of how PacBio sequencing works, should not be included in the ms.

20. L114, Initially…..

21. L114 should it be SOAPdenova or denovo?

22. L136 by the total number of codons

23. L143 and in the results (L263). What do these terms mean? What is the difference between a forward and a reverse repeat? Different strands? What is a complement repeat? This seems like it should be an inverted repeat?

24. L158 SNP arrays

25. L165 above was compared

26. L166 genome found and preliminary filtering performed ..

27. L170 SNP and is removed.

28. L175 1,000

29. L189 Figure and Table do not need underlining

30. L189 was 36.1% in both K….

31. L191 33.9% in both K…. (delete “respectively” in L192

32. L197 (Table 1). The non-coding…

33. Table 1 Fix brackets for (bp)

34. L207 genes in the K. also L210 genes in the K.  and also L212 genes in the K.    and L214 and L216

35. L227 are summarized in Table S1.

36. L230 in both the K.

37. L231. This information should not be included! There is ONLY ONE CODON for ATG Met and TGG Trp so it is obvious that there is no codon bias!!

38. L233 protein coding gene codons

39. Fig 2. The authors detail the aa frequency—but is this typical/standard or is there something different to note here?

40. L241. Please provide a reference for this statement of fact.

41. L248 regions rather than in IP…

42. L252 of the two

43. L262 results shown in

44. L261-271 Are the repeats in the same locations in the two genomes? Regulatory? Intergenic regions?

45. L279. This may be the first time that “cp” is used. It should probably be introduced as an abbreviation very early in the ms, rather than half way through.

46. L280 rpl22-rps19 genes (these are two genes)..were located

47. L83 genes were located

48. L287 by the ycf1 gene

49. L293 region, 256 bp away frm…

50. L310 using the mVISTA ..

51. L336 genomes of K….

52. L341. Which gene has a stop mutation? Is there a functional consequence? This information would be of interest to the reader.

53. L352 species, taxa were clustered into…

54. L369-371. Is this novel? Or expected for such cp genes? Please clarify in the text.

55. L384 and L385  highly

56. L393 for phylogenetic analysis, to resolve origin problems and ..

57. L395 in studies of phylogeny, evolution and species…

58. L397-403. This is mainly a duplication of data from the result section and should be streamlined.

59. L407 O. minuta which possesses a majority of

60. L411 relationship analysis

61. L422 should be rephrased for clarity

62. L423 What is a partial indel? This should be clarified for the reader

63. L425 from the wild…

Author Response

Dear reviewer,

       Thank you very much for your help. We revised the MS according to your comments.  Please find the attached file: Reponse to Reviewer 1.

Sincerely yours,

Dong-Mei Li

Guangdong Key Lab of Ornamental Plant Germplasm Innovation and Utilization, Environmental Horticulture Research Institute, Guangdong Academy of Agricultural Sciences, Guangzhou 510640, China

E-mail: [email protected]; Tel: +86 20 87593429; Fax: +86 20 87596402

Reviewer 2 Report

Manuscript ID: molecules-425498

Review Report

This report concerns the manuscript entitled “Complete Chloroplast Genome Sequences of Kaempferia galanga and Kaempferia elegans: Molecular Structures and Comparative Analysis”,authored by Dong-Mei Li, Chao-Yi Zhao, Xiao-Fei Liu.

This is an interesting study, presenting comprehensive data on the complete chloroplast genomes of two species of the genus Kaempferia. This study also includes comparisons with representatives of other genera from the Zingiberaceae.

Broad comments:

Besides an overall revision to make this manuscript more easily readable, the specific comments/suggestions below refer to aspects that, in my opinion, need reformulation.

Specific comments

Abstract

Lines 15, 21, 23: Please indicate full meaning of “LSC”, “SSC”, “IRs”, “SSRs”, “SNPs”.

This also applies to all first appearance of abbreviations along the manuscript.

Introduction

Line 41: “… is commonly cultivated…” NOT “… is common cultivated”

Line 42: “The morphological data…” NOT “The morphology data…”

Line 43: “… had been characterized by differences in….” NOT “… had been characterized differently by…”

Line 46: “…, to 10 cm, …”. What does this refer?

Lines 51 – 52: “… whereas K. elegans are mainly used as potted plant with ornamental value”. This sentence is not in accordance with what was said in lines 37 – 38, where it appears that both are used as ornamental and medicinal plants.

Moreover, the singular (“is”) should be used instead of the plural (“are”).

Lines 72 – 73: This sentence is out of the scope of this study: “…to discover the mechanisms of plant photosynthesis”. I suggest its removal.

Line 81: Please remove the unnecessary word “respectively”, since both genomes were subjected to both sequencing approaches.

Line 83: “…realized including …” NOT “… analyzed among…”.

Line 85: I suggest the reformulation of this last sentence, given the week meaning of the following suggestion: “…and will be beneficial for DNA molecular studies in Kaempferia.”.

Materials and Methods

Line 88 and Lines 88 - 90: “…, respectively, …” is an unnecessary word if both plants were cultivated in identical conditions and in the same place.

However, I do not understand the following sentence: “…, in greenhouse in environmental horticulture research institute, Guangdong academy of agricultural sciences, Guangzhou, China.”. Were both plants cultivated in the same place?

Line 91: Was the “sucrose gradient method” actually used to extract DNA? Please be more exact.

Line 92: Was “A260 and A280 used to measure DNA concentration of each sample”? Please be more accurate.

Lines 164 – 165: Please reformulate the following part of this sentence: “…, each chloroplast genome above….”

Results

Unnecessary word in each one of the following sentences:

Line 190: “…, respectively, …”

Line 192: “…, respectively, …” (the first one)

Line 251: “…, respectively)

Line 370: “…of…”

Line 195: “For…” NOT “With…”,

Lines 196 - 197: In the following sentence,respectivelymay be written in another position:

“… and third codons were respectively 55.4%, 62.6%, and 71.1% in K. galanga, respectively, and respectively 66.9%, 56.7%, and 64.7% in K. elegans, respectively (Table 1).

Line 222: “Lengths” NOT “Length”

Line 230: “both in …” NOT “in both…”. Please check, since this is repeated along the manuscript (for example in Lines 405 and 409 in the Discussion section).

Line 276: “… was performed for the four junctions …” NOT “… was performed of four junctions …”

Line 307: Suggestion: “More Higher divergence in the of coding regions was found…”

Line 313: “… on” NOT “… of”

Line 329: “Figure 7B showed that the average value of Pi was 0.01591…”. Is this value really shown in Figure 7B? Please reformulate this sentence.

Lines 331 – 333: There is a reference to seven highly divergent loci but I found out eight loci: rpL2-trnH is missing.

Lines 351 – 352: Suggestion: “The nine Zingiberaceae species clustered into four clusters. “ INSTEAD OF “Among nine Zingiberaceae species, they were clustered into four clusters.”

However, I would say that the representatives of the genera Kaempferia, Zingiber and Curcuma clustered together, though the two species of the genus Kaempferia constituted a separate and highly supported sub- clade. In a distinct node clustered together the representative of the genera Amomum and Alpina, where each genus also constituted separated and highly supported clades.

Line 367: Suggestion: Please indicate in a separate sentence that “no editing sites were found at the third codon position.”. This will clarify that this result respects to both species, not just to K. elegans.

Discussion

Line 377: Suggestion: “In this study, we obtained the sequences of the complete chloroplast…”

Line 390: Suggestion: “one in the IR regions.”

Line 394: “resolution origin problems, …”. What do you intend to say with this sentence?  Please clarify.

Line 395: “resources” NOT “resource”

Line 401: Unnecessary word: “were” (twice)

Line 403: Unnecessary word: “were

Line 404: “A. Kravanh”. Please indicate the full species name since it was not previously referred. Please check all equivalent situations.

Line 405: Unnecessary word: “type

Lines 407 – 408: I do not understand the following sentence: “… but were different from O. minuta with possessing the majority of dinucleotide repeat motif SSRs.” Please reformulate it.

Lines 413 – 417: Suggestion: “From the SNPs results, the number of 536 nucleotide substitutions is 536, which was were detected between K. galanga and K. elegans chloroplast genomes. It indicated, as shown below, that the more nucleotide substitution events occurred in the chloroplast genomes of the Kaempferia species were more than between species of Oryza, Machilus, cultivated Fagopyrum, Citrus and Panax, but less than between species of Solanum and wild Fagopyrum.”

Line 422: Suggestion: “Out of the 107 indels found between…”

Line 426: Suggestion: “the wild potato species…”

Line 427: Suggestion: “These indels and SNPs found in”

Line 429: Suggestion: “well as they were detected the 65 indels detected between M. yunnanensis and M. balansae.

Line 437: “… two Kaempferia species from” NOT “… two Kaempferia species in”

Lines 438 – 441: These two sentences need reformulation, according with the comments presented in the Results section.

Kaempferia species clustered together, with bootstrap values of 100%, as well as Amomum and Alpinia species, which segregated in two sister clades.

Line 453: Suggestion: “… chloroplast genomes were also calculated for the two Kaempferia species with and related Zingiberaceae species.”

Author Response

Dear reviewer,

     Thank you very much for your help. Please find the attached file: Reponse to Reviewer 2. We revised the MS according to your comments.

Sincerely yours,

Dong-Mei Li

Guangdong Key Lab of Ornamental Plant Germplasm Innovation and Utilization, Environmental Horticulture Research Institute, Guangdong Academy of Agricultural Sciences, Guangzhou 510640, China

E-mail: [email protected]; Tel: +86 20 87593429; Fax: +86 20 87596402

Reviewer 3 Report

Overall a well conducted study - most of my comments are editorial and can be noted on the attached (annotated) pdf file of the manuscript. The addition of a few brief comments on introns (Introduction and Results) might be useful, cpDNA typically has group II  introns (ORF less) and of course the trnK - matK encoding group II intron.  The description of the PACbio sequencing protocol is a bit long and the background information on PABbio can probably be removed or condensed. 

Author Response

Dear reviewer,

     Thank you very much for your help. Please find the attached file: reponse to reviewer 3. We revised the MS according to your comments. 

Sincerely yours,

Dong-Mei Li

Guangdong Key Lab of Ornamental Plant Germplasm Innovation and Utilization, Environmental Horticulture Research Institute, Guangdong Academy of Agricultural Sciences, Guangzhou 510640, China

E-mail: [email protected]; Tel: +86 20 87593429; Fax: +86 20 87596402
